# *Helicobacter pylori* infection and eradication outcomes among Vietnamese patients in the same households: Findings from a non-randomized study

**Long Van Dao**[1,2], **Hang Viet Dao**[1,2]*, **Hao Thi Nguyen**[1], **Vung Thi Vu**[1], **Anh Thi Ngoc Tran**[1], **Vu Quoc Dat**[3], **Long Bao Hoang**[1], **Hong Thi Van Nguyen**[1,2], **Thang Duy Nguyen**[1]

1 Institute of Gastroenterology and Hepatology, Hanoi, Vietnam, 2 Internal Medicine Faculty, Hanoi Medical University, Hanoi, Vietnam, 3 Department of Infectious Diseases, Hanoi Medical University, Hanoi, Vietnam

* daoviethang@hmu.edu.vn

**Data Availability Statement:** Data used in this study are available at: https://dataverse.harvard.edu/dataset.xhtml?persistentId=doi:10.7910/DVN/92FCEW.

## Abstract

### Objective

Familial transmission can possibly influence the infection and treatment of *Helicobacter pylori*. This study aimed to describe the prevalence of *H. pylori* infection and outcomes of eradication treatment among Vietnamese patients who live in the same households.

### Methods

We conducted a prospective cohort study of Vietnamese household members with upper gastrointestinal complaints. Participants received esophagogastroduodenoscopy and *H. pylori* testing. The *H. pylori*-positive patients were treated and asked to return for follow-up within 4 months. To explore factors associated with *H. pylori* infection at baseline, we performed multilevel logistic regression to account for the clustering effect of living in the same households. To explore factors associated with eradication failure, we used Poisson regression with robust variance estimation to estimate the risk ratio.

### Results

The prevalence of *H. pylori* infection was 83.5% and highest among children <12 years old (92.2%) in 1,272 patients from 482 households. There were variations in *H. pylori* infection across households (intraclass correlation = 0.14, 95% confidence interval (CI) 0.05, 0.33). Children aged <12 years had higher odds of *H. pylori* infection (odds ratio = 3.41, 95%CI 2.11, 5.50). At follow-up, *H. pylori* was eradicated in 264 of 341 patients (77.4%). The risk of eradication failure was lower for the sequential regimen with tetracycline.

### Conclusion

*H. pylori* infection was common among people living in the same households. Eradication success for *H. pylori* was higher for the tetracycline sequential regimen. More research

**Funding:** The authors received no specific funding for this work.

**Competing interests:** The authors have declared that no competing interests exist.

should be focused on how family factors influence *H. pylori* infection and on eradication treatment.

## Introduction

*Helicobacter pylori* is an important cause of gastric cancer [1], gastritis, peptic ulcer, and mucosa-associated lymphoid tissue lymphoma [2]. The prevalence of *H. pylori* infection ranges from 62.6% to 77.7% in low- to middle-income countries and as high as 24.4% in high-income countries [3, 4]. In Vietnam, *H. pylori* was found in 55.5%–65.5% of patients with upper gastrointestinal symptoms [5, 6] and in 38%–75% of the general population [7–9]. In addition to environmental and sanitary factors, behavioral factors, such as feeding children chewed food and consuming raw vegetables can cause intrafamilial transmission that can increase the risk of *H. pylori* infection [10–15].

   *H. pylori* eradication has been a global challenge due to widespread antibiotic resistance. Primary and secondary resistance to clarithromycin, metronidazole, and levofloxacin has been reported widely in some countries [16]. In Vietnam, the prevalence of resistance to antibiotics for *H. pylori* eradication has increased in recent years, and resistance to some antibiotics could reach >60% [9]. Familial factors might also limit eradication. The "Whole family-based *H. pylori* eradication strategy" has been proposed to acknowledge the role of these factors to achieve better and sustainable eradication success, but the strategy remains controversial [17].

   Although the evidence of *H. pylori* infection and antibiotic resistance in Vietnam is fairly abundant, no studies have focused particularly on members in the same households. The study aim was to estimate the prevalence of *H. pylori* infection among people living in the same households and their outcomes of eradication treatment.

## Methods

### Study population

A prospective cohort study was conducted at the Institute of Gastroenterology and Hepatology, Hanoi, Vietnam, from November 2017 to October 2019. First, we screened all patients (adults and children aged >3 years) with upper gastrointestinal symptoms (epigastric pain, heartburn, and indigestion) who tested positive for *H. pylori*. If these patients met the selection criteria and agreed to participate in the study, we also invited their household members to participate in the study. The exclusion criteria for the patients included pregnant and breastfeeding women, patients on proton-pump inhibitors, antibiotics or bismuth in the past 4 weeks, and patients who had neuropsychiatric problems. Finally, we recruited a total of 1,272 eligible participants in 482 households (63.1% of the total number of household members).

### Study procedures

Baseline demographics (e.g., sex, age, number of members in the household) and clinical data were collected in all participants. Participants who were *H. pylori*-positive were prescribed one of the four eradication regimens (see ***H. pylori* treatment**) and were asked to return for follow-up after 8 to 16 weeks, including testing for *H. pylori* again. All participants finished the study procedures after this testing. Positive-test participants continued to be evaluated and managed per routine practice. A flow diagram summarizing the number of participants and households for each stage of the study is shown in Fig 1.

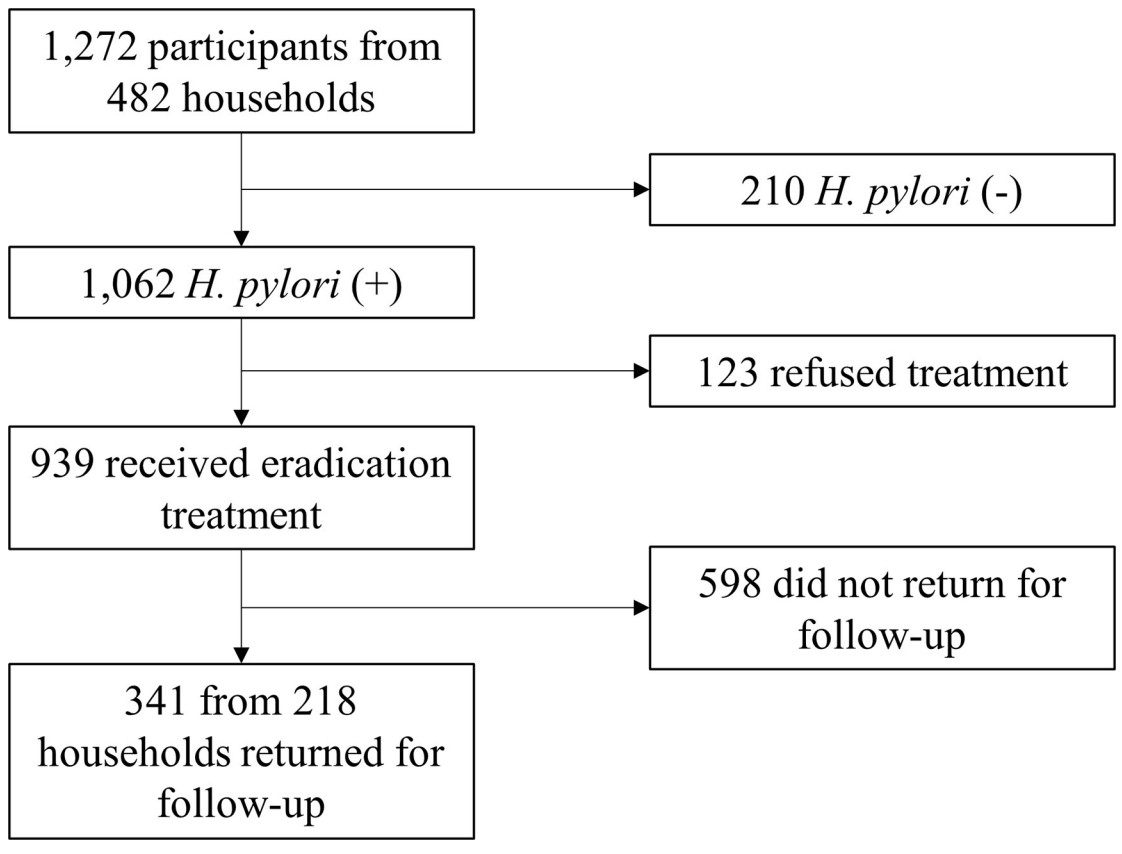

**Fig 1. Study flow diagram.**

### *H. pylori* diagnosis

Rapid Urease Test (RUT) is the most commonly used biopsy-based method in esophagogastro-duodenoscopy (EGD) for diagnosing *H. pylori* infection due to its low cost and simplicity. The test gives the best results if it is done on two biopsy samples—one collected from the antrum and the other from the gastric body [18]. Urease Breath Test (UBT) is a non-invasive method based on the ability of *H. pylori* to convert urea to ammonia and carbon dioxide. The version that uses the nonradioactive carbon-13 isotope is safe for children and women of childbearing age [19].

In our study, for adults, *H. pylori* was first detected by the RUT following the Maastricht V/ Florence Consensus Report [20]. If the RUT was negative but endoscopic findings suggested *H. pylori* infection or the patient had a history of peptic ulcer, we performed the UBT. If the UBT was positive, the patient was considered *H. pylori* positive. In children, we tested for *H. pylori* when they presented with symptoms of peptic diseases or iron-deficiency anemia of which other etiologies had been excluded. In most cases, pediatric EGD with anesthesia was performed, and *H. pylori* infection was confirmed by the RUT or UBT if patients could cooperate well.

The household members in the study were screened for peptic symptoms. If they reported upper gastrointestinal symptoms, we asked them to undergo EGD. Patients who underwent EGD were tested by the RUT for *H. pylori*. In other cases, only the UBT was performed.

### *H. pylori* treatment

Positive participants received an eradication regimen based on age, medical history, weight, and previous treatments. These regimens included two 14-day sequential therapies, a bismuth

quadruple therapy, and a triple therapy with levofloxacin. The detailed regimens are as follows: (1) clarithromycin sequential regimen—esomeprazole, amoxicillin, and bismuth for the first 7 days, followed by esomeprazole, clarithromycin, metronidazole, and bismuth for the second 7 days; (2) tetracycline sequential regimen—esomeprazole, amoxicillin, and bismuth for the first 7 days, followed by esomeprazole, metronidazole, tetracycline, and bismuth for the second 7 days; (3) bismuth quad regimen—esomeprazole, amoxicillin, metronidazole and bismuth for 14 days; and (4) levofloxacin regimen—esomeprazole, amoxicillin, and levofloxacin for 10 days. The bismuth quad and levofloxacin regimen were used if patient had a history of failure with other regimens.

## Outcomes

The primary outcome was *H. pylori*-positivity at baseline, and the secondary outcome was eradication failure at the follow-up visit after 8 weeks within 4 months. Although distinguishing between eradication failure and re-infection was difficult due to the short follow-up time, we assumed that positivity for *H. pylori* at follow-up was due to eradication failure. Since the treatment regimens were not randomized and were chosen by the examining doctor, we did not calculate sample size and power for the secondary outcome.

## Data analysis

The participants' characteristics are presented as numbers (percentages), means (standard deviations, SDs), or medians (interquartile range, IQR). Differences in characteristics between the *H. pylori*-positive and -negative participants were determined by performing the chi-square test, *t*-test, or Wilcoxon rank-sum test, where appropriate.

To explore the effect of being in the same household, we used random-effects logistic regression models with a random intercept of household (the cluster variable). In addition to adjusting for the clustering effect (i.e., members within a household might not have independent outcomes), these models also estimated the intraclass correlation (ICC), which shows how much variation is contributed by the clustering effect. For the primary outcome, we fitted two different models: model 1 included age (as a categorical variable) and sex as covariates, and model 2 included family membership as the only covariate. For the secondary outcome, we fitted a model, including age, sex, and eradication regimen. As in simple logistic regression, the adjusted estimate in the multilevel logistic regression model was the odds ratio (OR).

If the random-effects model did not provide a better fit (i.e., likelihood ratio test vs. simple logistic regression was not significant, which is the case of the secondary outcome analysis), we ran a simple regression model. The choice of regression models depended on the outcome prevalence. In our study, the secondary outcome was common (prevalence >15%). Therefore, we chose log-binomial regression or Poisson regression with robust variance estimation to account for the inflation of OR that is commonly observed in common outcomes [21]. Since our log-binomial model failed to converge, Poisson regression with robust variance estimation was used to estimate the adjusted risk ratio (RR). The model for the secondary outcome included age, sex, and eradication regimen, as in the multilevel model.

A p-value of <0.05 was considered to be indicative of statistical significance. All statistical analyses were performed in Stata 16/MP software (College Station, Texas, USA).

## Ethical approval

Informed consent was obtained from all adult participants. For children under 18 years of age, informed consent was obtained from their legal representatives (usually parents);

**Table 1. Comparison of the participants' characteristics at baseline between those who tested negative and positive for *H. pylori* (n = 1,272).**

| Characteristics, n (%) | Total | *H. pylori* negative (n = 210) | *H. pylori* positive (n = 1,062) | p-value[a] |
|---|---|---|---|---|
| **Sex** | | | | 0.41 |
| Female | 788 (53.5) | 107 (15.7) | 574 (84.3) | |
| Male | 591 (46.5) | 103 (17.4) | 488 (82.6) | |
| **Age (years)** | | | | <0.001 |
| <12 | 334 (26.3) | 26 (7.8) | 308 (92.2) | |
| 12–18 | 135 (10.6) | 21 (15.6) | 114 (84.4) | |
| >18–<45 | 600 (47.2) | 122 (20.3) | 478 (79.7) | |
| ≥45 | 203 (15.9) | 41 (20.2) | 162 (79.8) | |
| **Family membership** | | | | <0.001 |
| Father | 324 (25.5) | 77 (23.8) | 247 (76.2) | |
| Mother | 430 (33.8) | 76 (17.7) | 354 (82.3) | |
| Son | 267 (21.0) | 26 (9.7) | 241 (90.3) | |
| Daughter | 251 (19.7) | 31 (12.4) | 220 (87.6) | |

[a]Differences were tested by Chi-square test.

children >12 years also signed an assent form. The study was approved by the ethics committee of the Dinh Tien Hoang Institute of Medicine (approval No. 04/DTHIM-IRB).

## Results

We recruited 1,272 participants from 482 households (2,016 members in total), among which 53.5% were women, and the most common ages were in the range of 18–45 years (47.2%). Most households (84%) had 2–3 members participating in our study. The prevalence of *H. pylori* was 83.5% (1,062 participants). The prevalence of *H. pylori* was highest among children <12 years old (92.2%) and sons (90.3%) (Table 1).

In the multilevel models for the primary outcome (*H. pylori* positive), there was a variation in the outcome among households (likelihood ratio test p = 0.02; ICC = 0.14, 95% confidence interval (CI) 0.05, 0.33). This finding indicated that between-cluster variation (caused by living in the same households) contributed to 14% of the total variation. In the model that adjusted for age and sex, children aged <12 years had higher odds of being *H. pylori* positive than adults aged >18 to <45 years (OR = 3.41, 95%CI 2.11, 5.50). All other memberships (mother, son, and daughter) had higher odds of being *H. pylori* positivity than the fathers (Table 2).

Only 939 out of 1,062 positive participants decided to receive eradication treatment; the most common regimens were the tetracycline sequential (79.1%) and clarithromycin sequential (13.7%) regimens. Among these participants, 341 (36.3%) from 218 households returned for follow-up within 4 months. More than half (56%) of the households had only one member who returned for follow-up. The participants who returned for follow-up did not differ in terms of age and sex, but more participants returned for follow-up among those who received the clarithromycin sequential regimen (48.8%) than among those who received the tetracycline sequential regimen (34.3%) and bismuth quad regimen (36.7%) (chi-square p = 0.009) (S1 Table).

Among 341 participants who returned for follow-up, 77 (22.6%) remained positive for *H. pylori*. On univariate analysis, participants who had eradication success and failure did not differ in age, sex, and family membership (Table 3), but there were significantly more failures in the clarithromycin regimen (39.7%).

**Table 2. Differences in *H. pylori* infection at baseline (n = 1,272).**

| Covariate | OR (95%CI) | |
|---|---|---|
| | **Model 1 (adjusted for sex and age)** | **Model 2 (adjusted for family membership)** |
| **Male** | 0.79 (0.58, 1.09) | - |
| **Age** (reference: >18–<45 years) | | |
| <12 | 3.41 (2.12, 5.50) | - |
| 12–18 | 1.56 (0.90, 2.71) | - |
| ≥45 | 1.06 (0.69, 1.65) | - |
| **Family membership** (reference: father) | | |
| Mother | - | **1.50 (1.03, 2.19)** |
| Son | - | **3.10 (1.87, 5.14)** |
| Daughter | - | **2.45 (1.50, 3.99)** |
| **Random effects** | | |
| ICC | 0.14 (0.05, 0.33) | 0.14 (0.05, 0.32) |

OR, odds ratio; CI, confidence interval; ICC, intraclass correlation. Odds ratios (95%CI) in bold are statistically significant.

**Table 3. Characteristics of *H. pylori*-treated participants who returned for follow-up within 4 months, stratified by eradication status (n = 341).**

| Characteristic, n (%) | Total | Success (n = 264) | Failure (n = 77) | p-value[a] |
|---|---|---|---|---|
| **Sex** | | | | 0.075 |
| Female | 194 (56.9) | 157 (80.9) | 37 (19.1) | |
| Male | 147 (43.1) | 107 (72.8) | 40 (27.2) | |
| **Age (years)** | | | | 0.51 |
| <12 | 97 (28.4) | 70 (72.2) | 27 (27.8) | |
| 12–18 | 34 (10.0) | 27 (79.4) | 7 (20.6) | |
| >18 to <45 | 151 (44.3) | 119 (78.8) | 32 (21.2) | |
| ≥45 | 59 (17.3) | 48 (81.4) | 11 (18.6) | |
| **Family membership** | | | | 0.18 |
| Father | 78 (22.9) | 60 (76.9) | 18 (23.1) | |
| Mother | 122 (35.8) | 98 (80.3) | 24 (19.7) | |
| Son | 69 (20.2) | 47 (68.1) | 22 (31.9) | |
| Daughter | 72 (21.1) | 59 (81.9) | 13 (18.1) | |
| **Regimen** | | | | 0.003 |
| Clarithromycin sequential[b] | 63 (18.5) | 38 (60.3) | 25 (39.7) | |
| Tetracycline sequential[c] | 255 (74.8) | 209 (82.0) | 46 (18.0) | |
| Bismuth quad[d] | 32 (9.4) | 16 (72.7) | 6 (27.3) | |
| Levofloxacin[e] | 1 (0.3) | 1 (100) | 0 (0.00) | |

[a]Differences were tested by Chi-square test.

[b]Clarithromycin sequential regimen: esomeprazole, amoxicillin, and bismuth for the first 7 days, esomeprazole, clarithromycin, metronidazole, and bismuth for the second 7 days.

[c]Tetracycline sequential regimen: esomeprazole, amoxicillin, and bismuth for the first 7 days, esomeprazole, metronidazole, tetracycline, and bismuth for the second 7 days.

[d]Bismuth quad regimen: esomeprazole, amoxicillin, metronidazole, and bismuth for 14 days.

[e]Levofloxacin regimen: esomeprazole, amoxicillin, and levofloxacin for 10 days.

**Table 4. Factors associated with *H. pylori* eradication failure (N = 341).**

| Covariate | RR (95%CI) |
|---|---|
| **Male** | 1.46 (0.99–2.14) |
| **Age (years)** (reference: >18–<45) | |
| <12 | 0.46 (0.22–0.94) |
| 12–18 | 0.68 (0.35–1.33) |
| ≥45 | 0.74 (0.41–1.32) |
| **Regimen**[a] (reference: Tetracycline[b]) | |
| Clarithromycin sequential[c] | **3.72 (2.03–6.79)** |
| Bismuth quad[d] | **2.57 (1.07–6.18)** |

RR, risk ratio; CI, confidence interval. Risk ratios (95%CI) in bold are statistically significant.

[a]Levofloxacin regimen was excluded due to small sample size (n = 1).

[b]Tetracycline sequential regimen: esomeprazole, amoxicillin, and bismuth for the first 7 days, esomeprazole, metronidazole, tetracycline, and bismuth for the second 7 days.

[c]Clarithromycin sequential regimen: esomeprazole, amoxicillin, and bismuth for the first 7 days, esomeprazole, clarithromycin, metronidazole, and bismuth for the second 7 days.

[d]Bismuth quad regimen: esomeprazole, amoxicillin, metronidazole, and bismuth for 14 days.

The random-effects model (excluding one participant on the levofloxacin regimen) did not significantly improve the model fit (likelihood ratio test p = 0.19); therefore, a Poisson regression model with robust variance estimation was employed. In this model, children aged <12 had a lower risk of failure to eradicate than adults aged from 18 to 45 years (RR = 0.46, 95%CI 0.22, 0.95). Compared with the tetracycline sequential regimen, the clarithromycin (RR = 3.72, 95%CI 2.04, 6.80) and bismuth quad regimens (RR = 2.57, 95%CI 1.07, 6.19) had higher risks of failure to eradicate (Table 4).

## Discussion

In this prospective study in 1,272 participants from 482 households, *H. pylori* was detected in 83.5% of the baseline population. There were variations in the *H. pylori* infection rates among household members at baseline. Eradication success after 4 months was observed in 77.4% of the follow-up population and significantly higher in patients prescribed tetracycline sequential regimens.

Our estimate of *H. pylori* prevalence was higher than those reported in other countries, such as Thailand (43.6%) and China (32.6%) [3]. Some previous community-based studies in Vietnam have reported a wide range of prevalence from 34.1% to 79.4% [7–9], and a hospital-based study reported a prevalence of 66% [6]. There are several explanations for this variation. First, instead of estimating the prevalence from a study population randomly sampled from the general population, we recruited people who lived in the same households. The prevalence of infection in such a population might be higher if familial transmission was present and contributed to increased risks of *H. pylori* infection. Our study suggested some clustering effect of living in the same households on the prevalence of *H. pylori*; however, larger studies are needed to confirm this. Second, we could only invite 63% of members in the households to participate in the study. There might be differences between people who did and did not participate in the study; for example, one might be more likely to agree if they had some symptoms, and the prevalence of *H. pylori* might differ between symptomatic and asymptomatic individuals. Other factors that also contribute to the differences among studies include sensitivity/specificity of laboratory tests and differences in population characteristics.

Children aged <12 years in our study, regardless of sex, had higher odds of *H. pylori* infection. This result is consistent with previous findings in Vietnam, where children in the Highland area [22] and Northern rural area [8] had higher prevalence than those of adults [23]. However, a 2014 study in China showed that children aged >12 (34.8%) had a higher prevalence of *H. pylori* infection than children aged <12 years [24]. Despite variation in the prevalence trends among different pediatric age groups, the findings suggest that intrafamilial transmission might be a factor of *H. pylori* infection in children [25–27]. These findings might be because children have more exposure to sources of infection, such as being fed chewed food by the caretaker [28]. Therefore, measures to contribute to *H. pylori* eradication must involve the whole family and include different strategies, such as pharmacologic interventions, hygiene control, and food safety [17].

About 20% of patients who returned remained positive after treatment, an estimate that was similar to estimates from previous studies in Northern Vietnam (2%–32%) [29–31]. Antibiotics resistance or treatment non-adherence could explain the *H. pylori* treatment failure. Prior studies have found that the rate of antibiotics resistance of *H. pylori* was increasing in Vietnam, especially to triple therapy, which used to be the first line in *H. pylori* eradication [32–34]. This high eradication failure might also be due to the effect of living in the same household (e.g., other members influenced treatment adherence or shared an antibiotic-resistant strain). We did not find a clustering effect in eradication failure, which is probably attributable to the small number of people who returned for follow-up and to the many households with only one member. A study with a larger sample size and better control of loss-to-follow-up might allow us to explore the clustering effect more appropriately.

All regimes in our study were administered for a period of 14 days per the American Journal of Gastroenterology Clinical Guideline [35]. The tetracycline sequential regimen was most common (74.8%). Compared with the tetracycline regimen, the clarithromycin and bismuth quad regimens had a higher risk of failure to eradicate. Findings from some earlier studies in Vietnam also demonstrated similar trends; for example, the primary resistance rates of clarithromycin (34.1%) and metronidazole (27.9%) were higher than that of tetracycline (17.9%) [9]. Also, *H. pylori* eradication by using triple therapy as the first-line regimen significantly decreased from 91.0% in 2000 [36] to 62.8% in 2011 [37]; hence, a sequential or concomitant or bismuth-containing quadruple therapy has been highly recommended for patients [9].

There were some important limitations in this study that should be considered. First, our study might be subject to selection bias because we could not recruit every member in the households, and many patients did not return for follow-up, which could have biased our estimates of *H. pylori* infection and eradication failure. That limitation might also have reduced the power of the multilevel analysis because some clusters had few individuals. Loss-to-follow-up might result from patient's attitude towards *H. pylori* infection, that it is not a serious condition. In our experience, *H. pylori*-infected patients often do not come back for follow-up if their symptoms are resolved. Second, the non-standard regimens used at our institute and the lack of adherence data rendered the eradication failure findings more difficult to interpret. However, the strong associations between eradication failure and certain regimens might suggest an actual difference in treatment efficacy among *H. pylori* antibiotics.

## Conclusions

The prevalence of *H. pylori* infection among people living in the same households was high, particularly in children <12 years. Eradication success for *H. pylori* was higher for the tetracycline sequential regimen than for the other regimens. More research should be focus on how family factors influence *H. pylori* infection and on eradication treatment.

## Supporting information

**S1 Table. Characteristics of *H. pylori*-treated patients who did and did not return for follow-up within 4 months.**
(DOCX)

## Acknowledgments

The authors gratefully thank the board of directors and staff at Hoang Long General Clinic for supporting us in patient recruitment and data collection. We also thank all patients who were willing to participate in this study.

## Author Contributions

**Conceptualization:** Long Van Dao, Hang Viet Dao, Vung Thi Vu, Long Bao Hoang, Thang Duy Nguyen.

**Data curation:** Long Van Dao, Hang Viet Dao, Hao Thi Nguyen, Vung Thi Vu, Anh Thi Ngoc Tran, Vu Quoc Dat.

**Formal analysis:** Long Van Dao, Hang Viet Dao, Anh Thi Ngoc Tran, Long Bao Hoang.

**Investigation:** Hao Thi Nguyen, Vung Thi Vu.

**Methodology:** Long Van Dao, Hang Viet Dao, Vung Thi Vu, Vu Quoc Dat, Hong Thi Van Nguyen.

**Project administration:** Hao Thi Nguyen, Vung Thi Vu.

**Supervision:** Long Van Dao, Hang Viet Dao, Hong Thi Van Nguyen, Thang Duy Nguyen.

**Validation:** Hang Viet Dao.

**Writing – original draft:** Long Van Dao, Hang Viet Dao, Hao Thi Nguyen, Vu Quoc Dat.

**Writing – review & editing:** Long Van Dao, Hang Viet Dao, Anh Thi Ngoc Tran, Long Bao Hoang, Hong Thi Van Nguyen, Thang Duy Nguyen.

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
