## [Decision Letter · Decision Letter 0]

20 Aug 2021

PONE-D-21-20149

Helicobacter pylori infection and eradication outcomes among Vietnamese patients in the same households

PLOS ONE

Dear Dr. Dao,

Thank you for submitting your manuscript to PLOS ONE. After careful consideration, we feel that it has merit but does not fully meet PLOS ONE’s publication criteria as it currently stands. Therefore, we invite you to submit a revised version of the manuscript that addresses the points raised during the review process.

Clarifications are needed on methodology and low rate of followup. 

We look forward to receiving your revised manuscript.

Kind regards,

Iddya Karunasagar

Academic Editor

PLOS ONE

4. Thank you for stating the following in the Funding Section of your manuscript:

“This study was financially supported by the Hoang Long General Clinic, Hanoi, Vietnam”

“NO - The funders had no role in study design, data collection and analysis, decision to publish, or preparation of the manuscript.”

6. Please note that in order to use the direct billing option the corresponding author must be affiliated with the chosen institute. Please either amend your manuscript to change the affiliation or corresponding author, or email us at plosone@plos.org with a request to remove this option.

Additional Editor Comments (if provided):

The reviewer has raised number of points that need clarifications. Since there were 4 treatment regimes, how the patients were distributed among these groups. It is not clear if randomisation was done. Explanation for low rate of followup is needed and conclusions in abstract and text need to be harmonised

Reviewers' comments:

Reviewer's Responses to Questions

**Comments to the Author**

1. Is the manuscript technically sound, and do the data support the conclusions?

Reviewer #1: Partly

2. Has the statistical analysis been performed appropriately and rigorously? 

Reviewer #1: I Don't Know

3. Have the authors made all data underlying the findings in their manuscript fully available?

Reviewer #1: Yes

4. Is the manuscript presented in an intelligible fashion and written in standard English?

Reviewer #1: No

5. Review Comments to the Author

Reviewer #1: My comments to the article are as follows

1. The study is important and addresses an important research question regarding the antibiotic eradication of h.pylori among the familial contacts of h.pylori patients

2. There are 4 h.pylori eradication regimens. How were patients distributed among the 4 regimens? Was any randomization done? Was the distribution blind ? How was the number in each group arrived at? Any NNT done ? Is the study powered enough to analyse the differences in h.pylori eradication rates of different regimens?

3. We see that levofloxacin based regimens was second line treatment but had only 1 patient. Does it mean there was only one h.pylori eradication failure? If not what happened to others? What were the outcomes in them?

4. Non-adherence to antibiotic protocols. It would have easily found how much non-adherence to antibiotic protocol resulted in failure to treatment. Was anything done to improve follow up. What is experience of other articles/ authors regarding this issue of follow up?

5.Conclusions mentioned in the abstract and in the final paper do not match.

6.follow up patients is very less ~33% (341/939). Authors have not offered any explanation for the same.

The study main and important issue was familial transmission regarding h.pylori. Unfortunately, the conclusion say nothing about this point.

6. PLOS authors have the option to publish the peer review history of their article (what does this mean?). If published, this will include your full peer review and any attached files.

Reviewer #1: No

---

## [Author Response · Author response to Decision Letter 0]

7 Sep 2021

We would like to thank the reviewer for their insightful comments on our manuscript. Please find below our responses to the comments.

1. The study is important and addresses an important research question regarding the antibiotic eradication of H. pylori among the familial contacts of H. pylori patients.

Thank you.

2. There are 4 H. pylori eradication regimens. How were patients distributed among the 4 regimens? Was any randomization done? Was the distribution blind? How was the number in each group arrived at? Any NNT done? Is the study powered enough to analyse the differences in H. pylori eradication rates of different regimens?

This was not a randomized controlled trial. Since our primary outcome was the prevalence of H. pylori infection among people living in the same households, we conducted a cohort study and did not plan on randomizing the treatment regimens. Treatment was decided by the examining doctor (clarithromycin and tetracycline regimens were first-line regimens, and bismuth quad and levofloxacine regimens were second-line). Therefore, we did not calculate sample size and power for the secondary outcome (treatment outcome).

3. We see that levofloxacin-based regimen was second line treatment but had only 1 patient. Does it mean there was only one H. pylori eradication failure? If not, what happened to others? What were the outcomes in them?

We have discussed with the doctors who treated the study patients. Some also used the bismuth quad regimen for patients with history of failure to eradicate. Therefore, we have revised our manuscript to reflect this (see Lines 112-113).

4. Non-adherence to antibiotic protocols. It would have easily found how much non-adherence to antibiotic protocol resulted in failure to treatment. Was anything done to improve follow up. What is experience of other articles/authors regarding this issue of follow up?

Non-adherence is important but very difficult to resolve. Although H. pylori infection is common, it is not considered a serious condition in Vietnam, and many patients do not present with severe symptoms. Therefore, they may miss some doses, stop the treatment as soon as they feel better, and decide not to return for follow-up. As described in our study, only more than 1/3 of the patients who received treatment returned for follow-up.

In a clinical trial on H. pylori that we have just finished, we followed up study participants after 2 weeks since enrollment. A substantial portion of patients forgot some doses during the 2 weeks of therapy (we couldn't share the specific number since it's not published), and some did not agree to return for follow-up. Our strategy was to explain to them how adherence to H. pylori treatment is important (emphasizing the risk of antibiotic resistance), but the practice did not improve in many patients. We think a broader approach must be sought, for example, mass communication through television and online channels.

5. Conclusions mentioned in the abstract and in the final paper do not match.

We have revised both the conclusions in the abstract and full text.

6. Follow up patients is very less ~33% (341/939). Authors have not offered any explanation for the same.

We mentioned the significant loss-to-follow-up rate in the paragraph about limitations in the Discussion section. We have expanded the discussion on this issue in the paragraph (see Lines 276–279).

The study main and important issue was familial transmission regarding H. pylori. Unfortunately, the conclusion says nothing about this point.

To attribute the higher prevalence among people in the same households to familial transmission, we will need a longitudinal study where the entire households are sampled from the general populations and followed up over time. We deemed our findings inadequate to conclude anything on familial transmission; therefore, we only stated that the prevalence among people who live in the same households was high (and from our discussion, higher than previously reported numbers).

---

## [Decision Letter · Decision Letter 1]

25 Oct 2021

PONE-D-21-20149R1Helicobacter pylori infection and eradication outcomes among Vietnamese patients in the same householdsPLOS ONE

Dear Dr. Dao,

Thank you for submitting your manuscript to PLOS ONE. After careful consideration, we feel that it has merit but does not fully meet PLOS ONE’s publication criteria as it currently stands. Therefore, we invite you to submit a revised version of the manuscript that addresses the points raised during the review process.

Please address two minor comments of the reviewer and provide justifications as needed.

We look forward to receiving your revised manuscript.

Kind regards,

Iddya Karunasagar

Academic Editor

PLOS ONE

Journal Requirements:

Additional Editor Comments:

Please see two minor comments from the reviewer.

Reviewers' comments:

Reviewer's Responses to Questions

**Comments to the Author**

1. If the authors have adequately addressed your comments raised in a previous round of review and you feel that this manuscript is now acceptable for publication, you may indicate that here to bypass the “Comments to the Author” section, enter your conflict of interest statement in the “Confidential to Editor” section, and submit your "Accept" recommendation.

Reviewer #1: All comments have been addressed

2. Is the manuscript technically sound, and do the data support the conclusions?

Reviewer #1: Yes

3. Has the statistical analysis been performed appropriately and rigorously? 

Reviewer #1: Yes

4. Have the authors made all data underlying the findings in their manuscript fully available?

Reviewer #1: Yes

5. Is the manuscript presented in an intelligible fashion and written in standard English?

Reviewer #1: Yes

6. Review Comments to the Author

Reviewer #1: The Authors have answered my questions honestly and to the best of their ability. I accept and am thankful to them for the same.

My Comments:

1. The title should be changed suitably to reflect that this is not a randomised study.

2. As the authors accepted that they have not calculated NNT and it may not be adequately powered study to study the differences between the different h.pylori treatment regimens, the same should be mentioned in the methods

the manuscript may be accepted after incorporating the above changes

7. PLOS authors have the option to publish the peer review history of their article (what does this mean?). If published, this will include your full peer review and any attached files.

Reviewer #1: No

---

## [Author Response · Author response to Decision Letter 1]

27 Oct 2021

1. The title should be changed suitably to reflect that this is not a randomised study.

We have added a few words to clariy that this is a non-randomized study.

2. As the authors accepted that they have not calculated NNT and it may not be adequately powered study to study the differences between the different H. pylori treatment regimens, the same should be mentioned in the methods the manuscript may be accepted after incorporating the above changes.

We have added a sentence to clarify that no sample size and power calculation was done for the treatment outcomes (see the Revised Manuscript, lines 114-116).

---

## [Editor Report · Decision Letter 2]

10 Nov 2021

Helicobacter pylori infection and eradication outcomes among Vietnamese patients in the same households: findings from a non-randomized study

PONE-D-21-20149R2

Dear Dr. Dao,

We’re pleased to inform you that your manuscript has been judged scientifically suitable for publication and will be formally accepted for publication once it meets all outstanding technical requirements.

Kind regards,

Iddya Karunasagar

Academic Editor

PLOS ONE

Additional Editor Comments (optional):

Changes made are satisfactory.
---

## [Editor Report · Acceptance letter]

12 Nov 2021

PONE-D-21-20149R2 

*Helicobacter pylori* infection and eradication outcomes among Vietnamese patients in the same households: findings from a non-randomized study 

Dear Dr. Dao:

I'm pleased to inform you that your manuscript has been deemed suitable for publication in PLOS ONE. Congratulations! Your manuscript is now with our production department. 

Kind regards, 

on behalf of

Dr. Iddya Karunasagar 

Academic Editor

PLOS ONE